# NT-FDS—A Noise Tolerant Fall Detection System Using Deep Learning on Wearable Devices

**DOI:** 10.3390/s21062006

**Published:** 2021-03-12

**Authors:** Marvi Waheed, Hammad Afzal, Khawir Mehmood

**Affiliations:** Department of Computer Software Engineering, National University of Sciences and Technology (NUST), Islamabad 44000, Pakistan; marviwaheed@gmail.com (M.W.); khawir@mcs.edu.pk (K.M.)

**Keywords:** Fall Detection System (FDS), Recurrent Neural Networks (RNNs), Bidirectional Long Short-Term Memory (BiLSTM), Deep Learning (DL), Activities of daily Life (ADL), SisFall dataset, UP-Fall Detection dataset

## Abstract

Given the high prevalence and detrimental effects of unintentional falls in the elderly, fall detection has become a pertinent public concern. A Fall Detection System (FDS) gathers information from sensors to distinguish falls from routine activities in order to provide immediate medical assistance. Hence, the integrity of collected data becomes imperative. Presence of missing values in data, caused by unreliable data delivery, lossy sensors, local interference and synchronization disturbances and so forth, greatly hamper the credibility and usefulness of data making it unfit for reliable fall detection. This paper presents a noise tolerant FDS performing in presence of missing values in data. The work focuses on Deep Learning (DL) particularly Recurrent Neural Networks (RNNs) with an underlying Bidirectional Long Short-Term Memory (BiLSTM) stack to implement FDS based on wearable sensors. The proposed technique is evaluated on two publicly available datasets—SisFall and UP-Fall Detection. Our system produces an accuracy of 97.21% and 97.41%, sensitivity of 96.97% and 99.77% and specificity of 93.18% and 91.45% on SisFall and UP-Fall Detection respectively, thus outperforming the existing state of the art on these benchmark datasets. The resultant outcomes suggest that the ability of BiLSTM to retain long term dependencies from past and future make it an appropriate model choice to handle missing values for wearable fall detection systems.

## 1. Introduction

The Internet of Things (IoT) has established itself as an indispensable part of current age of user centric connectivity. It is an evolving paradigm that connects the diverse utilities around us to the Internet by making use of wireless/wired technologies. IoT can be viewed as an intelligent global network with inter-operable components consisting of self-configuring capabilities that connect billions of devices via the Internet, making it a highly heterogeneous ecosystem. According to the 2020 conceptual framework [1], the IoT can be expressed as a straightforward formula
(1)IoT=Services+Data+Networks+Sensors.

The fundamental principles of interconnection in IoT allow access to remote sensor data and control of complex physical environment from a distance that inherently permits efficient decision making, realistic automation, pragmatic productivity, greater wealth generation and enhanced public safety. The domain of IoT covers a widespread spectrum of daily life applications such as intelligent transportation, business intelligence and big data analytics, smart healthcare facilities, intelligent monitoring, positioning and navigation and smart logistics and so forth. The potential impact of IoT is predicted to bring forward surplus business opportunities and to expand the economic growth of IoT based services. According to a report published by McKinsey [2], the annual economic implications of IoT in 2025 would amount to $3.9 trillion to $11.1 trillion a year. About 41% of this market share is earned by IoT based healthcare services. In the recent years, the field of medicine and healthcare have greatly benefited from the evolutionary advancements in sensor technologies and IoT providing user friendly and efficient services to patients. So much so that the term “Internet of Medical Things (IoMT)” has been coined. These innovative developments include Human Activity Recognition (HAR) systems, smart movement detectors, fitness tracking, indigestible sensors, asset management systems, personalized emergency response systems, Fall Detection Systems (FDS), diagnostics, development of robust EHR systems and so forth.

The World Health Organization (WHO) defines a fall as an occurrence of a subject coming at rest with the ground level or lower. Unintentional falls can occur due to various reasons like accidental situations, performing high risk strenuous activities, subject surrounding factors like slippery floors, physical factors like loss of consciousness, tripping, poor balance, effects of wrong or overdose of medication and so forth. The impact and detrimental effects caused by a fall are dependent on the severity of the fall, age and physical well being of the subject experiencing the fall. For example, a standard fall will have varying effects on elderly patients as compared to younger audience. After road traffic injuries, unintentional falls related injuries make up for the major cause of death with an approximation of 650,000 fatal falls occurring each year. Falls are considered as a fatal threat to morbidity and mortality of elderly people. Around the world, mortality estimates are highest among elderly population of 60 years and above. Approximately 50% of injury-related hospital admissions are observed in the elderly of 65 or more. As a result, an estimated 40% of the injury-related deaths occur due to falls in the senior population [3].

The advancement in micro sensors integrated with microelectromechanical technology and IoT have paved way for research in mobile healthcare monitoring like FDSs. FDSs integrate the basics of an IoT network including wide scale streaming data, heterogeneity, correlation of space and time and highly noisy data management to promptly identify occurrence of falls and provide timely assistance by issuing alarm notifications to concerned healthcare providers. A typical FDS relies on information provided by context aware or wearable systems to distinguish between falls and Activities of Daily Life (ADL) [4]. Vision based context aware systems utilize computer vision [5], depth images [6], background separation [7], shape variation [8], 3D silhouette vertical distribution [9] for fall detection. Fall detection techniques relying on ambience based context aware systems utilize floor vibration patterns [10], sound height information [11], proximity sensors [12] and thermal imaging techniques [13]. Wireless wearable sensors play a fundamental part in an IoT environment to help digitize quantities of the physical world like temperature, pressure, humidity, acceleration and so forth. An alternative to external sensors such as Passive Infrared (PIR) sensors, thermographic cameras, proximity sensors and floor vibration detectors and so forth, body mounted wearable sensors are fastened to the body of subject of interest. These sensors, collecting important data related to the patient’s body movement make up for an efficient solution for fall detection with their low costs, weight, small size, low power usage and convenience in portability. Commonly used wearable sensors include accelerometer, heart rate sensor, gyroscope and magnetometer. Table 1 summarizes the strengths and weaknesses of the mentioned fall detection approaches.

A pressing challenge faced in sensor accumulated data is observance of missing values. Missingness in sensor data can appear due to unreliable data delivery, synchronization mismatch, local interference and lossy sensor devices and so forth. Presence of missing values complicate drawing insightful inferences from data, degrading the performance of FDS leading to inaccurate, faulty outcomes. Dealing with such type of insufficient, incomplete sensor data based monitoring system can prove fatal for patient health and safety. Such sensor failures can be overcome by adding analytical redundancies to replace faulty sensors with estimation methods. In this regard, various fault tolerant systems have been proposed such as Napolitano et al. [14], that employs a neural network based estimators to perform sensor failure detection and identification (SFDI) and sensor failure accommodation (SFA); ref. [15] presents a similar approach based on fully connected cascade neural network (FCC NN) architecture which relies on neuron by neuron learning algorithm for training of the model. Similarly, deep learning based methods such as specialized denoising auto-encoder have also been proposed [16]. A sequence-to-sequence missing values imputation (SSIM) novel architecture based on deep learning approaches is proposed using LSTM in [17]. Association rule mining based approaches have also been proposed to deal with missing values in sensor data streams [18]. The approaches mentioned in [19] study the multi-feature combinations for sensor based human activity recognition.

In this work, we handle fall detection as a sequence classification problem using deep learning techniques and propose a noise tolerant, wearable sensor based FDS working in presence of missing values. The research analyzes different mechanisms of existence of missing values in data which affect the data authenticity and the quality of quantitative results drawn from it. Three realistic cases of loss in sensor data are presented with varying proportions of multivariate missingness generated through Missing Completely at Random (MCAR) mechanism. The proposed technique uses stacked Bidirectional Long Short-Term Memory (BiLSTM) blocks in many-to-one configuration, trained on data from multimodal and unimodal sources, to detect falls. Comparison with deep learning techniques like Long Short-Term Memory units (LSTM), Gated Recurrent Units (GRU), Convolutional Neural Networks (CNN), Random Forest, Support Vector Machine (SVM) and K-Nearest-Neighbours (KNN) demonstrate the ability of the proposed NT-FDS to produce satisfactory results for fall detection in presence of missing values.

This paper is organized as follows—firstly, Section 2 covers a comprehensive literature review with existing solutions for fall detection and related works. Section 3 describes in detail the proposed fall detection mechanism while Section 4 reports the analytical results and performance evaluation of the proposed solution. Finally, Section 5 concludes the paper.

## 2. Related Work

This section presents a brief overview of existing work in the domain of Fall Detection using Wearable Sensors, Deep Learning based methods for Fall Detection and Methods to deal with the data loss in Wearable Sensors.

### 2.1. Fall Detection Using Wearable Sensors

Wearable sensor technology is unarguably the most employed method for reliable fall detection. An alternative to external sensors, body mounted wearable sensors are fastened to the body of subject of interest at different locations. Commonly used wearable sensors include accelerometer, heart rate sensor, gyroscope, magnetometer and so forth. Tri-axial accelerometer with X, Y and Z axes are used to determine the location of the body and its motion by determining the change in velocity. Fall detection relies on the sudden increase of negative acceleration caused by shift in orientation from upright to lying flat position. Lai et al. [20] proposed an integrated technique to detect fall incidents in the elderly as well as the joint sensing of the injured body part in case of fall. A fall is traced when the acceleration obtained by the tri-axial accelerometer exceeds the normal acceleration range significantly. The gathered information is wireless transferred to a computer for further analysis. Wang et al. [21] proposed a threshold reliant fall detection mechanism by accumulating data from various sensory devices. Combined data from accelerometer, cardiotachometer and smart sensors are used to approach a high detection accuracy of 97.5%. Casilari et al. [22] assess the effectiveness of machine learning techniques like Convolutional Neural Network (CNN) when applied to acceleration data for fall detection. The proposed technique reduces the preprocessing cost by allowing CNN to automatically extract features from complex sensor data rather than feeding custom-made features to the model.

Gyroscope sensors measure the angular velocity which is the change in rotational angle per unit of time. Most research approaches incorporate a combination of multiple sensors when using gyroscope to track angular velocity while few use gyroscopes exclusively for fall detection. Bourke et al. [23] proposed a threshold reliant algorithm for detecting fall events by using information from a bi-axial gyroscope sensor array. The approach sets thresholds for resultant values of trunk angle, angular velocity and angular acceleration each. In case of an event surpassing these set thresholds, alarms are triggered, and a fall is detected. The proposed system attains a robust 100% accuracy of distinguishing falls from ADL when data analysis is performed using MATLAB.

Approaches such as [24] used multiple sensors to detect falls where gyroscopes and accelerometer-derived posture information are used to reduce both false positives (e.g., sitting down fast) and false negatives (e.g., falling on stairs) with low computational costs and fast response to improve fall detection accuracy. Nyan et al. [25] studied body segment kinematics to detect falls in a body area network of wearable inertial sensors (3D accelerometers and 2D gyroscope). Martinez-Villaseñor et al. [26] proposed data accumulation through multimodal sensors to configure a fall detection system which works on different combination of sensors to identify falls. The deployed sensors include wearable sensors (tri-axis accelerometer, gyroscope and light intensity), an electroencephalograph helmet, infrared sensors, and cameras in lateral and front viewpoints. Machine learning methods including support vector machines (SVM), random forest (RF), multilayer perceptron (MLP) and k-nearest neighbors (KNN) are used for classification of falls and ADL.

Multi-sensor fusion is an approach to integrate information from different sensor sources to formulate a unified picture. In comparison to a single sensor method, multi-sensor fusion approach is set to produce robust measurements and accurate detection. Li et al. [27] propose a combination of wearable tri-axial accelerometer and context aware sensors like a depth camera and a micro-Doppler radar for fusion system-based fall detection. The fusion of mentioned heterogeneous sensors results in improvement of overall performance with overall classification accuracy increasing up to 91.3%. Pierleoni et al. [28] employ accelerometer, gyroscope, barometer and magnetometer to detect fusion based falls with a quaternion filter extracting acceleration relative to Earth’s frame. Thresholding standards, applied on several features like altitude, angular velocity and acceleration deliver a 100% sensitivity and a 99% specificity for fall detection.

Embedded sensors in the smartphones are recently used for fall detection instead of carrying additional body worn sensors. He et al. [29] discussed the challenges faced by conventional body worn sensors when detecting falls and propose a fall detecting mechanism by integrating Fisher’s discriminant ratio criterion and J3 criterion to create an algorithm for feature selection. The method utilizes built-in kinematic sensors for data accumulation. A hierarchical classifier used to classify human activities reaches an accuracy of 95.03%, proving the practicality of utilizing embedded smartphone sensors for fall detection.

### 2.2. Deep Learning Techniques for Wearable Sensors Based Fall Detection

Recently applications of Deep Learning, with their ability to deliver remarkable results in comparison to extracting features manually, are commonly used for HAR and fall detection. DL approaches can compute adequate features without systematic analysis and expert knowledge of the feature space. Such techniques allow stacking of hidden layers to extract highly abstract features and make better re-use of learned features.

Mauldin et al. [30] proposed an Android application called SmartFall which utilizes accelerometer data gathered from a smartwatch to identify incidents of fall. The proposed DL model based on RNN and CNN outperforms SVM and Naïve Bayes by spontaneously learning subtle features from unprocessed data. Musci et al. [31] proposed an online FDS based on the publicly available SisFall dataset using RNN. Preprocessed raw data is fed to stacked LSTM layers followed by a fully connected layer. Additional batch normalization and dropout layers configured with a weighted cross entropy loss function are inserted in the neural network. The model attains 97.16% and 94.14% accuracy for falls and ADLs respectively. Perejón et al. [32] proposed a real time fall detection using wearable accelerometers. Four different architectures using RNNs with underlying LSTM and Gated Recurrent Units (GRU) blocks were analyzed using the SisFall dataset at different frequencies. Torti et al. [33] propose an RNN based remote monitoring system suitable for an embedded implementation on a micro-controller unit. General formulas for power consumption, power computation and memory determination are presented and validated through implementation on a sensor development kit. Wang et al. [34] analyze various lightweight and shallow neural networks that require lesser storage and computational resources. The research concludes with a lightweight supervised convolutional neural network achieving 99.9% detection accuracy for resource constraint wearable sensors. Table 2 summarizes the recent works applying deep learning techniques on wearable sensor based fall detection.

### 2.3. Methods to Deal with Loss of Sensor Data

Napolitano et al. [14] proposed a fault tolerant system to cater for two specific requirements: sensor failure detection and identification (SFDI) and sensor failure accommodation (SFA) by employing online learning neural networks estimators. A decentralized neural network accommodates a sensor failure by supplying an estimate for the failed sensor. The authors in [15] use a similar approach based on fully connected cascade neural network (FCC NN) architecture which relies on neuron by neuron learning algorithm for training of the model. The proposed sensor failure detection, identification and accommodation (SFDIA) scheme achieves a 99% detection accuracy for failures in pitch, roll, and yaw rate gyro sensors of an airplane. Basic statistical investigation techniques like expectation maximization, maximum likelihood, mean/mode/median substitutions, multiple imputations, pairwise and list-wise deletions seem unfit for sensor data environments based on accounts of inefficiency and inconsideration of temporal factors. Recent approaches to deal with missing values in sensor data streams include estimation using freshness association rule mining (FARM) to relate intrinsic characteristics between sensors when imputing missing values [18].

Jaques et al. [16] proposed a deep learning technique using a specialized denoising autoencoder to cater for missing values observed in multimodal data. Comparison with techniques like principal components analysis (PCA) demonstrate the ability of multimodal autoencoder (MMAE) to predict the feature values from multiple missing modalities effectively. Zhang et al. [17] proposed a sequence-to-sequence missing values imputation (SSIM) novel architecture based on deep learning approaches. Their technique utilizes LSTM with a a variable-length sliding window algorithm, allowing SSIM to be used on smaller datasets as well. Hossein et al. [19] explored different feature combinations when working with missing values in sensor-based human activity recognition. The proposed feature-based approach uses mean, variance, skewness and kurtosis as statistical features along with Naïve Bayes and RF classifiers on the HASC benchmark dataset to produce satisfactory recognition results.

## 3. Proposed Framework: NT-FDS a Noise Tolerant Fault Detection System

The aim of this research is to propose an accurate and precise fall detection mechanism while handling the disturbances caused by missing values observed in wearable sensor data. The adopted method relies on a DL approach, treating fall detection as a sequence classification problem in time series data. The proposed methodology is shown in Figure 1.

### 3.1. Datasets

The presented research is performed on wearable sensor datasets that have following characteristics—public availability of raw data, detailed documentation of performed activities and incorporation of both ADLs and falls in the activities performed. The datasets mentioned in the Table 3 meet the mentioned premises. For the purpose of this research, datasets acquired through inbuilt smartphone sensors were neglected. An underlying drawback of using smartphones for detecting falls could be the loose connection between the device and the subject’s body depending upon the placement of the smartphone. This could lead to the invalidity of produced results. Among the mentioned datasets in Table 3, two have been selected as the most appropriate choice: SisFall dataset and UP-Fall Detection dataset.

**SisFall** dataset is chosen since it contains the largest amount of data and heterogeneity in ADLs and subjects. The other datasets exclude the elderly population and have limited diversity in context of performed activities and number of subjects. This dataset is created by conducting a survey on 15 adults of 60 years of age and above for the psycho-physic program of the Universidad de Antioquia, and 17 retirement homes. 38 volunteers divided into two groups of young and elderly collaborated for the dataset. Details of each group are given in Table 4. Detailed descriptions of performed activities and snapshots of activities video shown in Table 5 and Figure 2 respectively. All these subjects performed 19 different ADLs and 15 categories of falls carried out over multiple trials.

The **UP-Fall** Detection dataset contains data from various sources: wearable sensors (tri-axial accelerometer and gyroscope, electroencephalograph headset), ambient luminosity sensors and context-aware infrared sensors. For the purpose of this research, we have chosen a subset of UP-Fall Detection dataset containing wearable tri-axial accelerometer and tri-axial gyroscope sensory values. The dataset includes standard sensor placement positions like waist, chest and foot as well as left wrist and thigh(pocket) simulating wearing a smart watch and placing a smart phone inn pocket, respectively. The subjects were made to perform 11 (5 falls and 6 ADL) different activities. Statistical summary of the subjects is given in Table 6. Each activity was performed for three trials by all the volunteered subjects. Falls were performed for a duration of 10 s whereas ADL had variable duration. The details of the performed activities are given in Table 7.

### 3.2. Preprocessing

Data preprocessing is a key step that reshapes data into desired clean formats that can be feasible for analysis later. The two datasets contain in-equal number of performed ADL and falls along with variable duration and number of trials per activity. This non-uniformity in the data can lead to biased learning during the training and validation phases resulting in inaccurate picture for identifying falls. In order to balance the collected data generated from each source, duration of execution for all the activities are analyzed. In case of SisFall dataset the minimum duration for an activity that is, (12 s) is chosen as the standard window size. For UP-Fall Detection dataset minimum duration of 10 seconds is used as the standard.

The next task carried out for the preprocessing of data is data annotation. The original SisFall dataset files contain tri-axial data from 3 sensors while the UP-Fall Detection dataset consists of tri-axial data from 5 sensors. The annotated versions of the datasets created for the selected subjects include the addition of activity and user ID labels in order to facilitate data analysis later. The classes include:**Fall**: this class characterizes the activity intervals when the subject suffers a dangerous state transition leading to a harmful shift of state, that is, a fall. All 15 types of falls performed by the participants are subsumed under the umbrella of this class label.**ADL**: this class characterizes the activity intervals when the subject maintains control of its state and performs tasks without abrupt state transitions which may lead to falls. All 19 types of ADLs performed by the participants are subsumed under the umbrella of this class label.

### 3.3. Missing Values

Missingness in data is defined as an absence of response from sensors or data collection sources where a response is expected. Missing values distort the meaningfulness of data and leads to data corruption. Hence, missingness in data is interpreted as noise in data. Missing values are characterized by their proportion, patterns of observance and mechanisms of existence in data. For understanding the concept of missing data mechanisms, the data matrix *X* is partitioned into incomplete subset with missing values as ($X_{missing}$) and observed subset with complete responses as ($X_{observed}$). Hence the dataset can be represented as:
(2)$$X=(X_{missing},X_{observed})$$

Let M be the matrix of missingness with the same dimensions as X. Matrix M consists of 1s and 0s only, with 1 corresponding to a value being observed and 0 to a value being missed. Let the distribution of M be given as P(M|Y,ξ), where
ξ=parameter of missingness.

The Missing Completely at Random (MCAR) mechanism in data is observed when the probability of existence of missing values in data does not rely on the observed responses ($X_{observed}$) or on the missing values that are expected to be obtained ($X_{missing}$) [46]. It is an ideal but unreasonable assumption which exists in cases like failure of equipment, technical unsatisfaction, loss of data in transferring and so forth. The distribution of M can be modeled as:
(3)$$P\left(M|X,\xi \right)=P\left(M|X_{observed},X_{missing},\xi \right)=P\left(M|\xi \right).$$

For the purpose of this research, MCAR missing data mechanism was considered with various proportions of missingness in the data. Experimental analysis is carried out by adding three different percentages; 20%, 30% and 40% of MCAR missingness in data.

### 3.4. Deep Learning Model

We treat fall detection as a sequence classification problem, classifying each incoming input sample into fall/ADL category. The choice of DL techniques takes into account the resource constraint limitations of the relatively cheap embedded sensor device especially in inference model. RNN with underlying stack of bidirectional LSTMs are the core of the propose model. Treatment of missing values like replacement with mean, median and so forth, imputation and prediction exhaust computational resources which are already limited for the sensing devices. Hence, treatment of missing values is out of scope of this research. We propose a method of detecting falls in the presence of missing values.

RNN is a class of artificial neural network derived from the feed forward networks [47]. However, the distinguishing feature between recurrent and feed-forward networks is presence of at least one feedback connection in the recurrent networks. This connection feeds a part of the produced output back to the input. Thus, the activations pass around in a loop enabling the network to learn sequences and perform temporal processing. Hidden state which serves as a memory, retaining sequential information that the network has witnessed so far from the preceding timestep, is the most noteworthy feature of RNN. The same function is applied to all the inputs (operating with the same set of parameters) and hidden layers to generate the output. This reduces the complexity of parameters, unlike other neural networks. The weight matrices act as filters deciding the significance assigned to the current and previous hidden states. The feedback loop very exists at every timestep and adds traces of past hidden states. Basic RNN architecture is shown in Figure 3. The RNNs do not suffer from the limitation of accepting fixed dimensions for inputs and outputs and are flexible to variable sized inputs and corresponding outputs. RNNs rely on an application of back-propagation when applied to sequence problems like that of time series data called back-propagation through time (BPTT). The BPTT algorithm provides sequential sets of input/output timesteps to the network. It unfolds the network and calculates inaccuracies across each timestep. Finally, the algorithm rolls up the network and updates weights across the network. This entire process is repetitive. The computational cost of BPTT has a direct relation with the number of timesteps. When the number of timesteps is higher, weight updates become an exhausting process depending upon calculations of derivatives. The weights eventually vanish or explode resulting in noisy model performance and poor learning. A gradient explains the change in all weights with regard to the change in error. Hence, the problem is identified as the vanishing or exploding gradient problem in RNNs [48].

Long Short-Term Memory networks or LSTMs are a special category of RNNs having the ability to learn long-term dependencies [49]. LSTMs are precisely conceived to cater the long-term dependency issue or the exploding/vanishing gradient problem resulting from BPTT. All RNNs contain repetitive modules of neural network. The chain like repeating structure in LSTMs is slightly different, consisting of four interactive neural network layers instead of one. The four gates include forget gate, input gate, output gate, and internal hidden state gate represented by f, i, o, g respectively. These four gates are utilized by LSTMs to perform a specific function of defining a cell state at each timestep. Details are appended or withdrawn from the cell state after careful regulation from these gates. The gates comprise a sigmoid neural net layer and a point-wise multiplication operation which output a zero or a one defining how much information to allow pass. Firstly, the forget gate regulates what to remove or forget in the cell state from the previous hidden state ($h_{t-1}$) information. Next, the information to be updated in the cell state is decided by the input gate layer. It is a two step procedure with a tanh layer creating a vector of new candidate values ($C^{\prime }t$), leading to the update of current cell state. Finally, a sigmoid layer filters out the required segment of the cell state to output. A succeeding tanh layer scales the output. The final output is yt. A vector set of equations describing the working of a typical LSTM are given as:
(4)$$f_{t}=\sigma \left(W_{f}\cdot \left[h_{t-1},x_{t}\right]+b_{f}\right)$$
(5)$$i_{t}=\sigma \left(W_{i}\cdot \left[h_{t-1},x_{t}\right]+b_{i}\right)$$
(6)$$C_{t}^{\prime }=tanh\left(W_{c}\cdot \left[h_{t-1},x_{t}\right]+b_{c}\right)$$
(7)$$C_{t}=\left(f_{t}*C_{t-1}\right)+\left(i_{t}*C_{t}^{\prime }\right)$$
(8)$$o_{t}=\sigma \left(W_{o}\cdot \left[h_{t-1},x_{t}\right]+b_{o}\right)$$
(9)$$h_{t}=o_{t}*tanh\left(C_{t}\right)$$
(10)$$y_{t}=W_{y}\cdot h_{t}.$$

In order to cater to the corruption and ambiguities caused by the missing data problem, we propose to utilize Bidirectional Long Short-Term Memory (BiLSTM) [50] blocks stacked on top of each other. BiLSTMs are a widely applied improvement of LSTMs which work better with sequence classification problems. Conventional LSTMs are only able to make use of the previous context but BiLSTMs process the data in both directions with two separate hidden layers. It’s like training two separate LSTMs on the input sequence given all the timesteps, one that is trained on the input sequence as is and the other on the reversed copy of the input sequence. The input is executed in two ways, one from past to future and the other from future to past.The outcomes of the two hidden layers are then fed forwards to the same output layer. This basic concept enables BiLSTMs to access long-range context in both input directions. BiLSTMs utilize both the previous and future context however, the forward pass and backward pass are completely independent of each other. The basic difference from a unidirectional LSTM is the ability of LSTM training in backwards direction to preserve information from the future as well. At any timestep the combined hidden layers preserve information from past and future. A typical BiLSTM model architecture is shown in Figure 4.

The overall proposed system design is shown in Figure 5. Two BiLSTM stacked on top of each other with additional dropout and fully connected layers are used. The dropout layers are only activated during the training phase and eliminated during the inference stage. 3D Data reshaping is done to make the dimensions of data in accordance with the LSTM parameter requirements. The three dimensions of input to each LSTM layer include samples, timesteps and features. The output dimensions are dependent upon the number of classes.

There are two approaches used in this research based on the number and kind of sensors used to accumulate the data. A multi-sensor fusion approach which uses the combination of sensors to gather data coming from different sources. This approach uses data from all three sensors of the customized embedded device (in case of SisFall dataset) and all five body worn sensors (in case of UP-Fall Detection dataset) to detect falls. While the single sensor approach chooses one sensor at a time to detect falls. This is done in order to understand the behavior of individual sensor data and how it contributes towards fall detection. The sensors considered for this approach are the ITG3200 gyroscope and the MMA8451Q accelerometer from the SisFall dataset and the wearable waist mounted IMU sensor from the UP-Fall dataset. All three percentages of missingness have been considered for the individual sensor case. The overall methodology, illustrated in Figure 5, is summarized in the form of algorithm in Algorithm 1.
**Algorithm 1:** Algorithm for Deep Learning based Missing Data Imputation and Fall Detection.**(1) Data Preprocessing:**Adjustment of activities duration, re-sampling and data selection.Data annotation into classes: FALL / ADL.**(2) Missing Data Pattern Generation:**Creation of MCAR pattern of missingness.Creation of noisy datasets with different percentages of MCAR missingness: 20%, 30%, 40%.**(3) Train/Test Split:**Division of datasets for training, testing and validation stages.**(4) 3D Data Reshaping:**Transformation of training data from 2D (samples, timesteps) to 3D (samples, timesteps, features) datasets.**(5) Deep Learning Based Fall Detection:**Creation of neural network based on stacked BiLSTMs, dropout layers, activation and fully connected layers.**(6) BiLSTM Network Training:**Hyper-parameters Optimization: choice of loss function, optimizer, hidden units, hidden layers, output layer with activations.Training of Network: Choice of batch size, early stopping percentage and epochs.**(7) BiLSTM Prediction:**Sequence Classification: Use of trained data to predict classes (FALL/ADL) during testing.**(8) Performance Evaluation:**Model loss and classification accuracy analysis.Confusion matrix creation.Effectiveness analysis: calculation of precision, sensitivity and specificity.

### 3.5. Experimental Setup

The proposed fall detection mechanism has been implemented using the Keras framework, a high-level framework for deep learning for Python programming language. All training procedures have been performed on LENOVO 80MK workstation, equipped with an Intel® Core™ i7-6500U CPU. The fall detection problem is treated as a sequence classification problem with fixed length sequence input using two BiLSTMs stacked on top of each other. Each BiLSTM layer contains 32 hidden neurons. Batch size is 2048 and dropout probability used in both layers is 0.2. Categorical cross entropy is chosen as the loss function. A softmax activation function is applied at the output for each input sequence. All BiLSTMs are trained using early stopping. A 90%/10% train/test split is used.

## 4. Performance Evaluation

Using classification accuracy alone as a performance measure can be confusing if there is an unequal number of observations in each class or if there are more than two classes in the dataset. Computing a confusion matrix gives better understanding of what the classification model is predicting correctly and the types of errors it is making. Most evaluation metrics are computed from the confusion matrix. It summarizes the number of correct and incorrect predictions with count values and divides them into each class. The rows depict actual classes while the columns represent the predicted class outcomes by the classifier. Some terms used to define a confusion matrix include:Positive (P): Observation is positive.Negative (N): Observation is not positive.True Positive (TP): Observation is positive. The prediction is positive.False Negative (FN): Observation is positive, but the prediction is negative.True Negative (TN): Observation is negative. The prediction is negative.False Positive (FP): Observation is negative, the prediction is positive.

A basic confusion matrix layout is shown in Figure 6. Details of evaluation metrics derived from the computed confusion matrix are given as follows.

Classification rate or accuracy is the ratio of error-free predictions made to the total number of predictions made by the classification model. However, this accuracy can be problematic since it includes both kinds of errors.
(11)Accuracy=(TP+TN)/(TP+TN+FP+FN).

Error rate is the ratio of all incorrect predictions to the total number of predictions made by the classification model.
(12)ErrorRate=(FP+FN)/(TP+TN+FP+FN).

Another informative measure is sensitivity or recall. Sensitivity is the ratio of number of correct positive predictions to the total number of positives.
(13)Sensitivity=TP/(TP+FN).

Specificity is the ratio of the number of correct negative predictions to the total number of negatives.
(14)Specificity=TN/(TN+FP).

Precision is the ratio of the number of error-free positive predictions to the total number of positive predictions. It is also called positive predictive value.
(15)Precision=TP/(TP+FP).

Table 8 describes the distribution of the two activities (ADL and falls) in training and testing datasets for the SisFall dataset.

Table 9 summarizes the distribution of activities for training and testing datasets for the UP-Fall Detection dataset.

### 4.1. Multisensor Fusion Approach

This section describes the experimental results of using a combination of sensors to detect falls. For experiments using the SisFall dataset, three tri-axial sensors (2 accelerometers and 1 gyroscope) working at a sampling frequency of 200 Hz are chosen to provide collected data from subjects executing 15 categories of falls and 19 different ADLs. For experiments using the UP-Fall dataset, five tri-axial sensors (each sensor consisting of an accelerometer and a gyroscope) working at the sampling frequency of 100 Hz are used to accumulate data for fall detection.

The experiment is performed on 4 different case scenarios: dataset with complete records and no observed missing values, incomplete dataset consisting of 80% of original data and 20% of the data observed as missing values through MCAR mechanism, incomplete dataset consisting of 70% of original data and 30% of the data observed as missing values through MCAR mechanism and incomplete dataset consisting of 60% of original data and 40% of the data observed as missing values through MCAR mechanism. Extensive hyper-parameters tuning and optimization were performed on a Keras framework using the Python programming language. Results obtained during training and testing are mentioned in Table 10 and Table 11 for the SisFall and UP-Fall datasets, respectively.

Figure 7 shows the confusion matrices for the four experimental case scenarios on the SisFall dataset, giving insights for the performance of the proposed deep learning based classification model. The confusion matrix for the ideal scenario of continuous streams of data with no interruptions and noisy missing values demonstrates error-free predictions for the case of ADL and only two misclassifications for FALL during testing, yielding a low error rate and perfect sensitivity. A trend in reduced performance and effectiveness analysis is observed as the percentage of missing values is increased in the data for the multi-sensor fusion approach. The analytical outcomes deduced from the confusion matrices are presented in the form of effectiveness analysis in Table 12.

The breakdown of true classification vs. predicted classification for the proposed multi-sensor fusion technique using UP-Fall Detection dataset is given in Figure 8. The confusion matrices for the four cases of missingness in data depict an indirect relation between the percentage of missing values in data and the correctly predicted classification. Increase in percentage of missingness in data led to decrease in correct predictions. For example, in the worst case scenario with 40% MCAR missingness in data 23.31% of actual falls are misidentified as ADL and 12.3% of true ADL are interpreted as falls. The analytical outcomes deduced from the confusion matrices are presented in the form of effectiveness analysis in Table 13.

### 4.2. The Single Sensor Approach

This section describes the experimental results of using individual sensors to detect falls. The single sensor approach chooses one sensor at a time to detect falls. This is done to analyze the behavior of individual sensor data and how it contributes towards fall detection.

For the SisFall dataset, the ITG3200 gyroscope and the MMA8451Q accelerometer are considered. Similarly, data from waist mounted tri-axial accelerometer and waist mounted tri-axial gyroscope is selected when using the UP-Fall dataset. All three percentages of missingness have been considered for the individual sensor case. The same hyper-parameters optimized for the combined sensor approach are applied when using the individual sensors. The resulting accuracy and loss for the single sensor approach using the accelerometer are given in Table 14 and Table 15 for SisFall and UP-Fall datasets respectively. Effectiveness analyses summarizing percentages of sensitivity, specificity, precision and error rate for the two datasets are given in Table 16 and Table 17.

The use of accelerometers for monitoring acceleration changes in three orthogonal directions for fall detection produces accurate results following the trends of the multi-sensor fusion approach. The technique on UP-Fall dataset produces more accurate results than on SisFall dataset. It also performs better than the multi-sensor fusion technique applied on the same dataset.

While accelerometers only have the capability of measuring linear motion, gyroscopes have the ability to measure the tilt and lateral orientation of an object. The accuracy results of using the ITG3200 gyroscope from the SisFall dataset and the waist mounted gyroscope from the UP-Fall dataset are given in Table 18 and Table 19 respectively. Effectiveness analyses for the two datasets are represented in Table 20 and Table 21.

In case of single sensor approach using gyroscope applied on SisFall dataset, the results for class predictions show reduced performance in terms of accuracy and effectiveness analysis. Out of 33 ADL during testing, only 25 are correctly identified, the remaining are misinterpreted as FALL for the ideal scenario with no missing values. Similarly 12 out of 44 FALL cases are misclassified and predicted as ADL. The worst case scenario with 40% MCAR missingness added to the data results in 32 out of 44 incorrect predictions for the class FALL and only 12 true class predictions during testing. The findings from the confusion matrices lead to escalated error rates and bleak percentage outcomes for sensitivity, precision and specificity.

The comparison between the combined sensors approach and single sensor approach when applied on SisFall dataset, indicates the superiority of the former where the combined sensor approach generates 97.4% accuracy when classifying falls and ADL for the complete dataset, 94.81% for the best-case scenario of missingness (i.e., 20%) and 88.31% for the worst (i.e., 40%). The effectiveness analysis supports the conclusion with 100%, 96.97% and 81.81% sensitivity for the three cases respectively. This could be justified as the multi-sensor fusion approach collects data from various sources which is favorable to overcome the limitations observed in one or more sensors.

In case of the proposed fall detection technique applied on the UP-Fall dataset, multi-sensor fusion approach produces acceptable results with 88% and 82.55% accuracy for the best-case and worst-case scenarios respectively. However, the best results in terms of accuracy and effectiveness analysis are achieved with the single sensor approach using accelerometer with a maximum accuracy of 97.21% and 99.77% sensitivity. This out-performance of the single sensor approach using accelerometer over the multi-sensor fusion approach could be justified by the variance of influential factors which contribute towards sensor data generation. Some of the contributing factors include but are not limited to: configurations of the sensing device (sampling rate, memory, range, power consumption and battery), categorisation and patterns of performed activities, characteristics of the volunteering subjects, experimental conditions for activity simulation and so forth. When comparing the performances of individual sensors in the single sensor approach, use of accelerometer outperforms that of gyroscope for both datasets. Results from generated losses and accuracies as well as the effectiveness analyses indicate the superiority of tri-axial accelerometers over gyroscopes to monitor inertial navigation for detecting falls.

### 4.3. Comparison with Existing State of the Art

Effectiveness comparison of proposed technique with recent fall detection methods are presented in Table 22 and Table 23. None of the mentioned literature cater to the problem of observing missing values in data for the used datasets. For the sake of comparison, we have used the best case scenarios with no missing values for the proposed fall detection technique.

## 5. Conclusions

Fall detection is a relevant public concern specially for the elderly and physically impaired. Recent years have witnessed considerable research on accurate fall detection based on wearable sensor technology. However, not many techniques focus on dealing with realistic, noisy, faulty or missing sensor data streams. The focus of this research is to apply DL techniques for wearable sensors based fall detection when faced with the predicament of observing missing values in data. Data acquisition from various multimodal and uni-modal sources is done to simulate real-life activities. The proposed technique applies RNN with an underlying stack of BiLSTM blocks on two publicly available datasets—SisFall and UP-Fall. Various proportions of MCAR missingness is generated and added to data to evaluate the performance of the proposed model under unfavourable, realistic and noisy conditions.

Fall detection is treated as a sequence classification problem by using a DL architecture. The ability of DL techniques to automatically extract features from complex training data make it an efficient application for real-life fall detection. BiLSTMs with their inherent ability to retain long-term dependencies from past and future, easily access the access the long-range context when a value in sensor data goes missing. This justification makes BiLSTM as an appropriate choice for our noise tolerant fall detection architecture on sequential sensor data. Two different approaches based on the number and kind of sensing device used to accumulate data are proposed. Multi-sensor fusion technique generally performs better for both datasets based on the ability of the proposed technique to overcome the limitations of one or more sensor source by relying on others. The comparison between individual sensors in the single sensor approach, promotes the performance of tri-axial accelerometer over gyroscope to navigate inertial changes causing falls. The results presented in this research conform to the effectiveness of using BiLSTM with its ability to learn two-way long-term dependencies in presence of missing values in data.

## Figures and Tables

**Figure 1 sensors-21-02006-f001:**
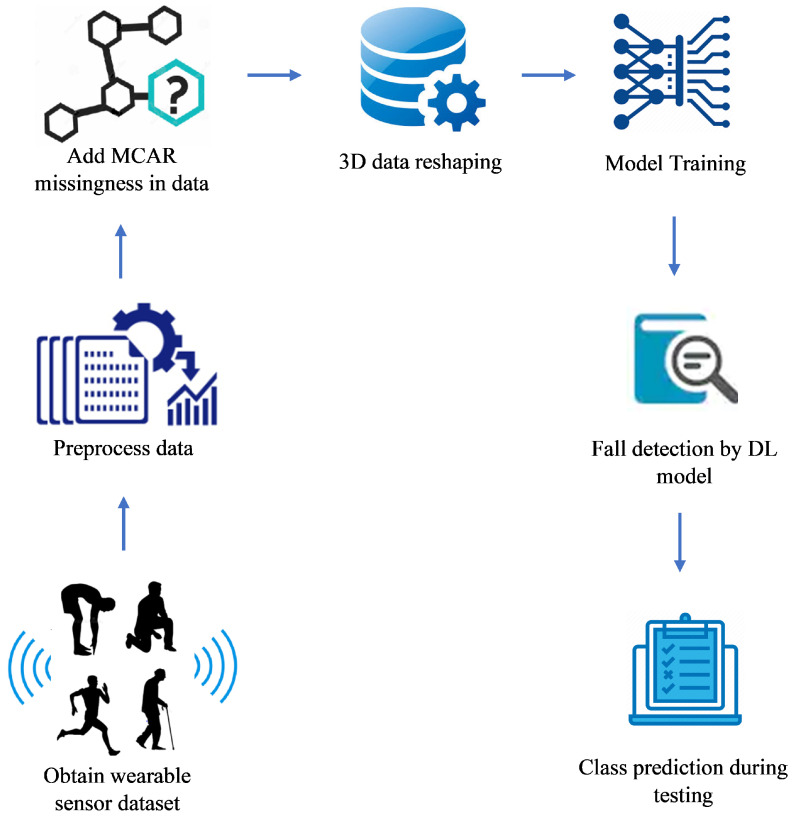
Proposed Framework: Noise Tolerant Fall Detection System (NT-DFS) A Noise Tolerant Fault Detection System.

**Figure 2 sensors-21-02006-f002:**
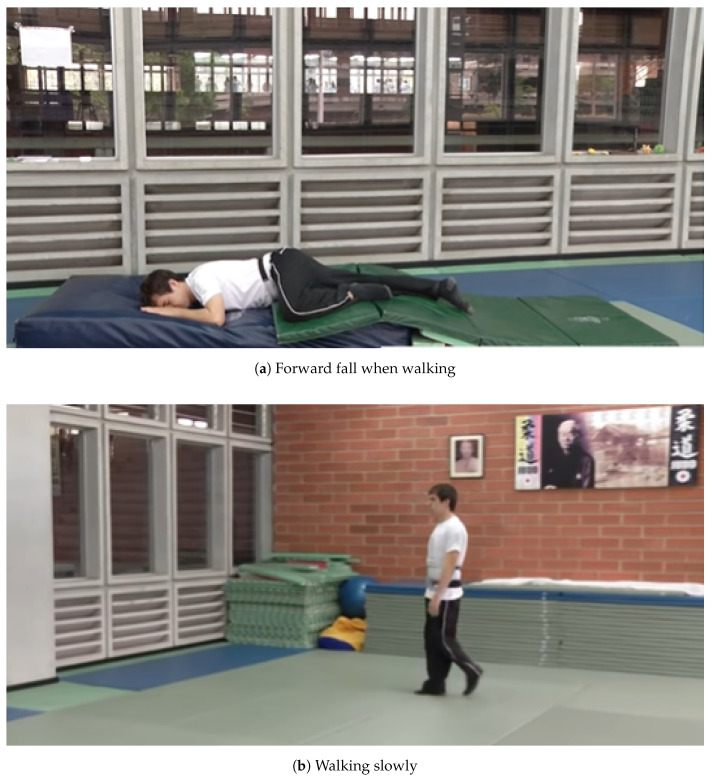
Snapshots from video footage of performed activities.

**Figure 3 sensors-21-02006-f003:**
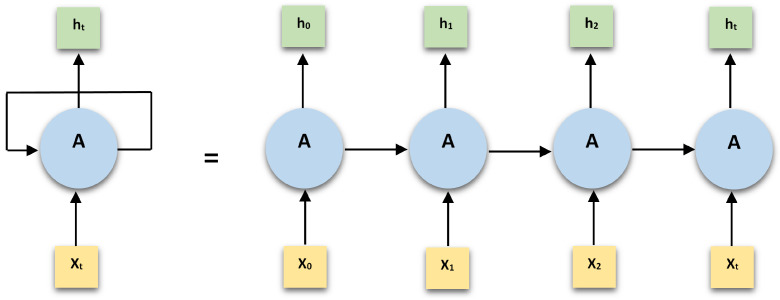
An uncoiled recurrent neural network (RNN).

**Figure 4 sensors-21-02006-f004:**
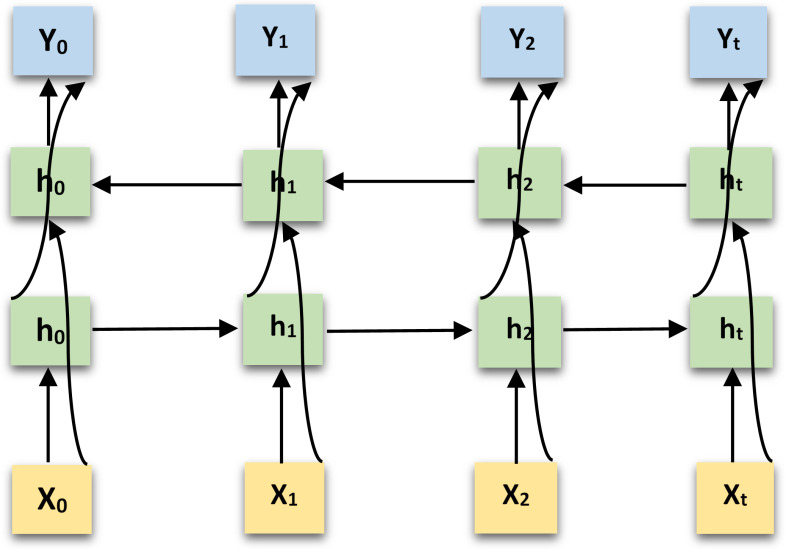
Bidirectional Long Short-Term Memory (LSTM) architecture.

**Figure 5 sensors-21-02006-f005:**
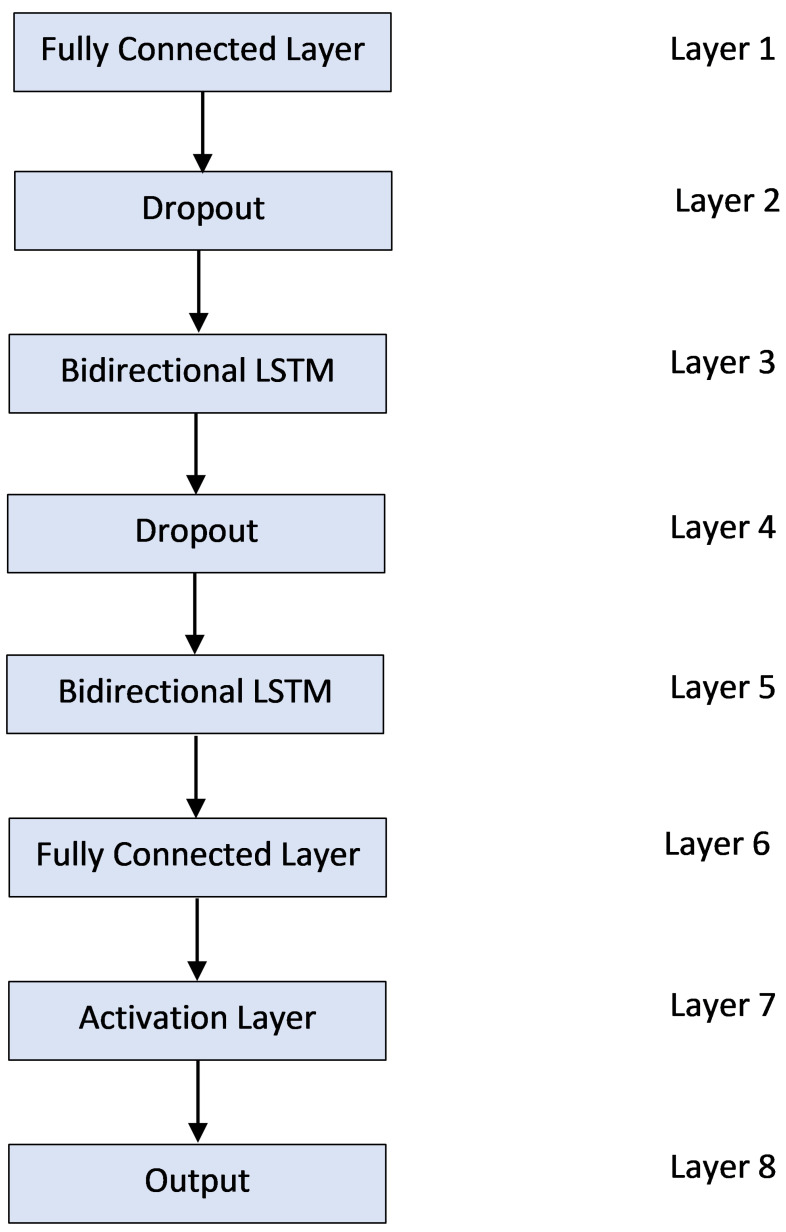
Overall design for the proposed DL based fall detection system.

**Figure 6 sensors-21-02006-f006:**
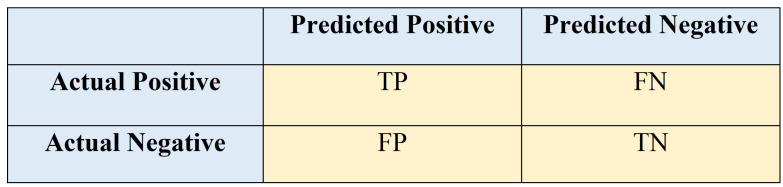
Confusion Matrix.

**Figure 7 sensors-21-02006-f007:**
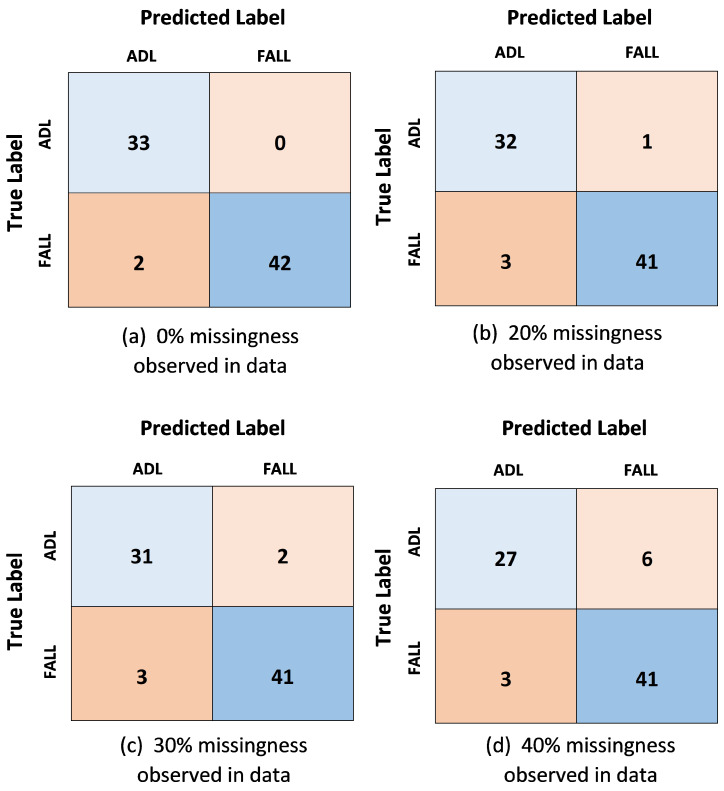
Confusion matrix resulting from testing the Multi-sensor Fusion approach using the SisFall dataset.

**Figure 8 sensors-21-02006-f008:**
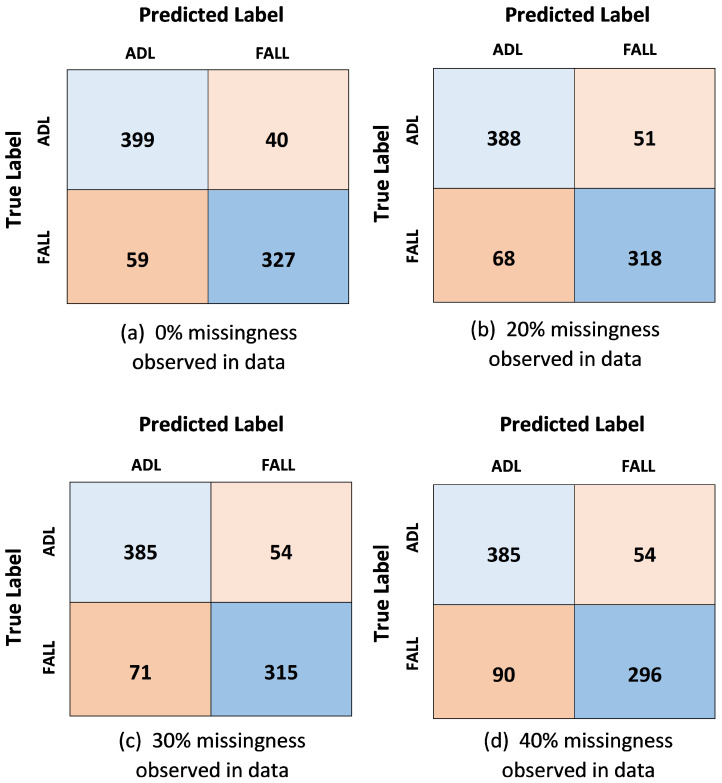
Confusion matrix resulting from testing the Multi-sensor Fusion approach using the UP-Fall dataset.

**Table 1 sensors-21-02006-t001:** Strengths and Weaknesses of Different Fall Detection Approaches.

Approach Used	Strengths	Weaknesses
Vision based fall detection	3D posture and scene analysis, inactivity monitoring, shape modeling, spatio-temporal motion analysis, occlusion sensitivity	Invasion of privacy, interference and noise in data, burdensome syncing of devices, difficult set up of devices
Ambience based fall detection	Safeguards privacy, robust occlusion sensitivity	Expensive equipment, detection dependent on short proximity range
Wearable sensors based fall detection	low costs, small size, light weight, low power consumption, portability, ease of use, protection of privacy, robust occlusion	Intrusive approach, sensors to be worn at all times
Sensor Fusion based fall detection	Robust measurements, accurate detection, high performance	Difficult set-up of equipment, complex syncing between devices
IoT based fall detection	High success rates for precision, accuracy and gain, accessibility with real-time patient monitoring	Threat of data security, compromise of privacy, strict global healthcare regulations

**Table 2 sensors-21-02006-t002:** Summary of most recent techniques for wearable fall detection using deep learning.

Ref.	Dataset	DL Algorithm Used	Accuracy (%)	Sensitivity (%)	Specificity (%)	Precision (%)
[30]	Smartwatch [35] Notch [36] Farseeing [37]	RNN (GRU)	85 99 99	100 80 55	70 99 99	77 79 37
[31]	SisFall [38]	CNN CAE CAE	99.94 99.91 99.81	98.71 99.2 99.07	99.96 99.93 99.83	NS
[30]	URFD [39]	CNN	99.86	99.72	100	100
[22]	UP-Fall [40]	CNN	75.89	96.08	59.02	NS

**Table 3 sensors-21-02006-t003:** List of publicly available datasets considered for Fall Detection System.

Dataset	No. of Subjects	Type of ADLs	Type of Falls	Sensing Device
MobiFall [41]	24 (22 to 42 years old)	9	4	Smartphone
tFall [42]	10 (20 to 42 years old)	7	8	Smartphone
Project gravity [43]	3 (ages 22, 26, and 32)	7	12	Smartphone
DLR [44]	16 (23 to 50 years old)	6	1	Wearable sensors
UMAfall [45]	17 (18 to 55 years old)	8	3	Wearable sensors
SisFall	23 (19 to 75 years old)	19	15	Wearable sensors
UP-Fall	17 (18 to 24 years old)	6	5	Multi-modal sensors (wearable, ambient and vision)

**Table 4 sensors-21-02006-t004:** Age, height, weight of the participating subjects in SisFall dataset.

	Age	Gender	No. of Subjects	Weight (kg)	Height (m)
Young Subjects	19–30	M	11	59–82	1.65–1.84
	19–30	F	12	41–64	1.50–1.69
Senior Subjects	60–71	M	8	56–103	1.63–1.71
	62–75	F	7	50–71	1.49–1.69

**Table 5 sensors-21-02006-t005:** List of activities (Falls and Activities of Daily Life (ADL)) performed in the SisFall dataset.

Activity Description	Act Code	Trial Period	Trials
Walking slowly	D01	100 s	1
Walking quickly	D02	100 s	1
Jogging slowly	D03	100 s	1
Jogging quickly	D04	100 s	1
Walking upstairs and downstairs slowly	D05	25 s	5
Walking upstairs and downstairs quickly	D06	25 s	5
Slowly sit in a half height chair, wait a moment, and up slowly	D07	12 s	5
Quickly sit in a half height chair, wait a moment, and up quickly	D08	12 s	5
Slowly sit in a low height chair, wait a moment, and up slowly	D09	12 s	5
Quickly sit in a low height chair, wait a moment, and up quickly	D10	12 s	5
Sitting a moment, trying to get up, and collapse into a chair	D11	12 s	5
Sitting a moment, lying slowly, wait a moment, and sit again	D12	12 s	5
Sitting a moment, lying quickly, wait a moment, and sit again	D13	12 s	5
Being on one’s back change to lateral position, wait a moment, and change to one’s back	D14	12 s	5
Standing, slowly bending at knees, and getting up	D15	12 s	5
Standing, slowly bending without bending knees, and getting up	D16	12 s	5
Standing, get into a car, remain seated and get out of the car	D17	12 s	5
Stumble while walking	D18	12 s	5
Gently jump without falling (trying to reach a high object)	D19	12 s	5
Falling forward when walking triggered by a slip	F01	15 s	5
Falling backwards when walking triggered by a slip	F02	15 s	5
Falling laterally when walking triggered by a slip	F03	15 s	5
Falling forward when walking triggered by a trip	F04	15 s	5
Falling forward when jogging triggered by a trip	F05	15 s	5
Falling Vertically when walking caused by fainting	F06	15 s	5
Falling when walking, with use of hands in a table to dampen fall, caused by fainting	F07	15 s	5
Falling forward while trying to get up	F08	15 s	5
Falling laterally while trying to get up	F09	15 s	5
Falling forward while sitting down	F10	15 s	5
Falling backwards while sitting down	F11	15 s	5
Falling laterally while sitting down	F12	15 s	5
Falling forward when sitting, triggered by fainting or falling asleep	F13	15 s	5
Falling backwards when sitting, triggered by fainting or falling asleep	F14	15 s	5
Falling laterally when sitting, triggered by fainting or falling asleep	F15	15 s	5

**Table 6 sensors-21-02006-t006:** Age, height, weight of the participating subjects in UP-Fall dataset.

Age	Gender	No. of Subjects	Weight (kg)	Height (m)
18–24	M	9	54–99	1.62–1.75
18–24	F	8	53–71	1.57–1.70

**Table 7 sensors-21-02006-t007:** List of activities (Falls and ADL) performed in the UP-Fall Detection dataset.

Activity Description	Act Code	Trial Period	Trials
Falling forward using hands	01	10 s	3
Falling forward using knees	02	10 s	3
Falling backwards	03	10 s	3
Falling sideward	04	10 s	3
Falling sitting in empty chair	05	10 s	3
Walking	06	60 s	3
Standing	07	60 s	3
Sitting	08	60 s	3
Picking up an object	09	10 s	3
Jumping	10	30 s	3
Laying	11	60 s	3

**Table 8 sensors-21-02006-t008:** Distribution of ADL and Falls in SisFall dataset.

	Train	Test
**ADLs**	362	33
**Falls**	331	44
**Total**	693	77

**Table 9 sensors-21-02006-t009:** Distribution of ADL and Falls in UP-Fall dataset.

	Train	Test
**ADLs**	4091	439
**Falls**	3334	386
**Total**	7425	825

**Table 10 sensors-21-02006-t010:** Results for Multi-sensor Fusion Approach using SisFall: Accuracy and Loss during training and testing phases.

Original Data Observed	MCAR Missing Values Observed	Training Accuracy (%)	Testing Accuracy (%)	Training Loss	Testing Loss
100	0	98.01	97.4	0.0749	0.1198
80	20	96.39	94.81	0.1002	0.107
70	30	95.85	93.5	0.1205	0.2259
60	40	88.81	88.31	0.2694	0.282

**Table 11 sensors-21-02006-t011:** Results for Multi-sensor Fusion Approach using UP-Fall: Accuracy and Loss during training and testing phases.

Original Data Observed	MCAR Missing Values Observed	Training Accuracy (%)	Testing Accuracy (%)	Training Loss	Testing Loss
100	0	89.51	88	0.2488	0.2899
80	20	86.70	85.58	0.2917	0.3179
70	30	85.10	84.85	0.2935	0.3321
60	40	83.6	82.55	0.3001	0.358

**Table 12 sensors-21-02006-t012:** Effectiveness Analysis for Multi-sensor Fusion Approach using SisFall dataset: Error rate, Sensitivity, Specificity & Precision during testing.

Original Data Observed	MCAR Missing Values Observed	Error Rate	Sensitivity (%)	Specificity (%)	Precision (%)
100	0	0.0259	100	95.45	94.28
80	20	0.0519	96.97	93.18	91.43
70	30	0.0649	93.93	93.18	91.17
60	40	0.1168	81.81	93.18	90

**Table 13 sensors-21-02006-t013:** Effectiveness Analysis for Multi-sensor Fusion Approach using UP-Fall dataset: Error rate, Sensitivity, Specificity & Precision during testing.

Original Data Observed	MCAR Missing Values Observed	Error Rate	Sensitivity (%)	Specificity (%)	Precision (%)
100	0	0.12	90.88	84.71	87.11
80	20	0.1442	88.38	82.38	85.08
70	30	0.1515	87.70	81.60	84.42
60	40	0.1745	87.70	76.68	81.05

**Table 14 sensors-21-02006-t014:** Results for Single Sensors Approach using Accelerometer for SisFall dataset: Accuracy and Loss during training and testing phases.

Original Data Observed	MCAR Missing Values Observed	Training Accuracy (%)	Testing Accuracy (%)	Training Loss	Testing Loss
100	0	97.65	96.1	0.084	0.1224
80	20	94.4	93.51	0.1571	0.196
70	30	87.73	87.01	0.3569	0.3765
60	40	82.67	81.82	0.4084	0.3827

**Table 15 sensors-21-02006-t015:** Results for Single Sensors Approach using Accelerometer for UP-Fall dataset: Accuracy and Loss during training and testing phases.

Original Data Observed	MCAR Missing Values Observed	Training Accuracy (%)	Testing Accuracy (%)	Training Loss	Testing Loss
100	0	95.3	97.21	0.1272	0.0841
80	20	94.11	95.88	0.1445	0.1026
70	30	90.79	93.82	0.2317	0.1521
60	40	88.32	91.39	0.282	0.2004

**Table 16 sensors-21-02006-t016:** Effectiveness Analysis for Single Sensor Approach using accelerometer on SisFall dataset.

Original Data Observed	MCAR Missing Values Observed	Error Rate	Sensitivity (%)	Specificity (%)	Precision (%)
100	0	0.039	100	93.18	91.67
80	20	0.065	96.97	90.90	88.89
70	30	0.13	87.87	86.36	82.85
60	40	0.181	84.84	79.54	75.67

**Table 17 sensors-21-02006-t017:** Effectiveness Analysis for Single Sensor Approach using accelerometer on UP-Fall dataset.

Original Data Observed	MCAR Missing Values Observed	Error Rate	Sensitivity (%)	Specificity (%)	Precision (%)
100	0	0.0278	99.54	94.56	95.41
80	20	0.0412	99.77	91.45	92.99
70	30	0.0618	98.86	88. 08	90.41
60	40	0.0860	99.77	81.86	86.22

**Table 18 sensors-21-02006-t018:** Results for Single Sensors Approach using Gyroscope on SisFall dataset: Accuracy and Loss during training and testing phases.

Original Data Observed	MCAR Missing Values Observed	Training Accuracy (%)	Testing Accuracy (%)	Training Loss	Testing Loss
100	0	77.62	74.03	0.4548	0.4754
80	20	70.04	66.23	0.5252	0.6135
70	30	64.80	62.34	0.6276	0.6207
60	40	57.76	46.75	0.6985	0.7331

**Table 19 sensors-21-02006-t019:** Results for Single Sensors Approach using Gyroscope on UP-Fall dataset: Accuracy and Loss during training and testing phases.

Original Data Observed	MCAR Missing Values Observed	Training Accuracy (%)	Testing Accuracy (%)	Training Loss	Testing Loss
100	0	79.93	78.55	0.4374	0.489
80	20	78.74	77.58	0.4508	0.49
70	30	75.88	74.79	0.4938	0.5226
60	40	72.73	72	0.54	0.552

**Table 20 sensors-21-02006-t020:** Effectiveness Analysis for Single Sensor Approach using gyroscope on SisFall dataset.

Original Data Observed	MCAR Missing Values Observed	Error Rate	Sensitivity (%)	Specificity (%)	Precision (%)
100	0	0.26	75.75	72.72	67.55
80	20	0.338	81.81	54.54	57.44
70	30	0.377	54.54	68.18	56.25
60	40	0.532	72.72	27.27	42.85

**Table 21 sensors-21-02006-t021:** Effectiveness Analysis for Single Sensor Approach using gyroscope on UP-Fall dataset.

Original Data Observed	MCAR Missing Values Observed	Error Rate	Sensitivity (%)	Specificity (%)	Precision (%)
100	0	0.2193	84.28	70.98	76.76
80	20	0.2242	82.68	71.76	76.90
70	30	0.2521	82.68	65.80	73.33
60	40	0.28	81.32	61.4	70.55

**Table 22 sensors-21-02006-t022:** Comparison of the the proposed NT-FDS on SisFall Dataset with State-of-the-art.

Ref	Dataset Used	DL Algorithm Used	Accuracy	Sensitivity (%)	Specificity (%)	Precision (%)
[31]	SisFall	RNN (LSTM)	97.16 (Falls) 94.14 (ADLs)	NS	NS	NS
[32]	SisFall	RNN (LSTM)	95.51	92.7	94.1	NS
[33]	SisFall	One Layer GRU Two Layer GRU One Layer LSTM Two Layer LSTM	96.4 96.7 96.3 96.1	88.2 87.5 88.2 90.2	96.3 96.8 96.4 97.1	68.2 68.1 69.5 68.3
**Proposed NT-FDS**	SisFall	BiLSTM	97.41	100	95.45	94.28

**Table 23 sensors-21-02006-t023:** Comparison of the the proposed NT-FDS on UP-Fall dataset with recent techniques.

Ref.	Dataset Used	DL Algorithm Used	Accuracy	Sensitivity (%)	Specificity (%)	Precision (%)
[51]	UP-Fall	2D CNN (vision based approach)	95.64	NS	NS	NS
[26]	UP-Fall	RF SVM MLP KNN	95.76 93.32 95.48 94.90	66.91 58.82 69.39 64.28	99.59 99.32 99.56 99.5	70.78 66.16 73.04 69.05
[22]	UP-Fall	CNN	75.89	96.08	59.02	NS
**Proposed NT-FDS**	UP-Fall	BiLSTM	97.21	99.54	94.56	95.41

## Data Availability

The experiments are performed on publicly available datasets. The sources for utilized datasets are available in [52,53].

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
