# Peer review of "NT-FDS—A Noise Tolerant Fall Detection System Using Deep Learning on Wearable Devices"

_sensors, 2021, doi:10.3390/s21062006_

Round 1

Reviewer 1 Report

The comments made to the authors of the article can be found inside the document.

Reviewer 2 Report

This is a very interesting study concerning the fall detection by using Deep Learning, simulating a scenario in which data are affected by noise. Noise has been simulated by introducing lacks of data in datasets.

Anyway, there are some issues, listed below, that need further investigations.

The paper can be considered for publication after major revision.

  • English needs to be revided throughout the whole manuscript;
  • The first part of the paper, in particular sections 2 and 3, are very heavy to read. Authors should make a big effort to summarize the sections to make reading more fluent;
  • Table 1 should propose a comparison between Fall Detection approaches. However, it describes only the vision based approach;
  • In Table 4 I suggest to introduce the number of each kind of participants also;
  • Please, explain better in the text the interpretation of Figures 7 and 9;
  • Why should this approach be better than the others in the literature? Authors should emphasize the effectiveness of the proposed procedure compared to the others.

Round 2

Reviewer 2 Report

The authors reviewed the article according to the reviewer's suggestions.

I consider the paper acceptable for publication.

However, the improvement of figures resolution is desirable, s well as a further check of the English.

Author Response

Comments by Reviewer: The authors reviewed the article according to the reviewer’s suggestions. I consider the paper acceptable for publication. However, the improvement of figures resolution is desirable, s well as a further check of the English.

Response: As suggested, the quality of figures has been improved by using high resolution pictures. Higher resolution images for Figures 1,3,4,5,6,7 and 8 are added.
